# Inhibition of VEGF receptors induces pituitary apoplexy: An experimental study in mice

Yoshito Sugita[1,2☯], Shigeki Takada[1,2☯], Kenji Tanigaki[2]*, Kazue Muraki[2], Munehiro Uemura[2], Masato Hojo[1,3]*, Susumu Miyamoto[1]

**1** Department of Neurosurgery, Kyoto University Graduate School of Medicine, Kyoto, Japan, **2** Research Institute, Shiga Medical Center, Shiga, Japan, **3** Department of Neurosurgery, Shiga General Hospital, Shiga, Japan

☯ These authors contributed equally to this work.
* mhojo7@gmail.com (MH); tanigaki@res.med.shiga-pref.jp (KT)

## Abstract

Anti-vascular endothelial growth factor (VEGF) therapy has been developed for the treatment of a variety of cancers. Although this therapy may be a promising alternative treatment for refractory pituitary adenomas and pituitary carcinomas, the effects of anti-VEGF agents on the pituitary gland are not yet well understood. Here, we found that mice administered with OSI-930, an inhibitor of receptor tyrosine kinases including VEGF receptor 1 and 2, frequently exhibited hemorrhage in the pituitary gland. This is the first report that anti-VEGF therapy can cause pituitary apoplexy. C57BL/6 mice were daily injected intraperitoneally with 100 mg/kg body weight of OSI-930 for one to six days. Pituitary glands were immunohistochemically examined. Four of six mice treated for three days and all of five mice treated for six days exhibited hemorrhage in the pituitary gland. In all cases, the hemorrhage occurred just around Rathke's cleft. In OSI-930-administered mice, the vascular coverage and branching were reduced in the anterior lobe, and capillary networks were also decreased in the intermediate lobe in a treatment-day dependent manner. Few blood vessels around Rathke's cleft of the intermediate lobe express VE-cadherin and are covered with platelet-derived growth factor receptor-β (PDGFR-β)-positive cells, which suggests that capillaries around Rathke's cleft of the intermediate lobe were VE-cadherin-negative and not covered with pericytes. The reduction of capillary plexus around Rathke's cleft was observed at the site where hemorrhage occurred, suggesting a causal relationship with the pathogenesis of pituitary hemorrhage. Our study demonstrates that anti-VEGF agents have a risk of pituitary apoplexy. Pituitary apoplexy should be kept in mind as an adverse effect of anti-VEGF therapy.

## Introduction

Anti-angiogenic therapy has been developed for the treatment of a variety of cancers. Vascular endothelial growth factor A (VEGF-A) is well known to be the main molecular driver of tumor angiogenesis [1]. Currently, bevacizumab, a recombinant monoclonal antibody

**Data Availability Statement:** All relevant data are within the paper and its Supporting Information files.

**Funding:** This work was partially supported by the Japan Society for the Promotion of Science (20K09360 to Masato Hojo), and by an institutional grant from Shiga Medical Center Research Institute (to Kenji Tanigaki). The funders had no role in study design, data collection and analysis, decision to publish, or preparation of the manuscript.

**Competing interests:** The authors have declared that no competing interests exist.

targeting VEGF-A, is widely used for the treatment of high-grade gliomas [2, 3]. Hemorrhage is one of the most common adverse events of anti-VEGF therapy for cancer [4, 5]. The mechanisms of hemorrhagic complications induced by anti-VEGF agents are complex and remain to be clarified. Inhibition of VEGF decreases the renewal capacity of endothelial cells when vessels are damaged, which might increase the risk of hemorrhage. However, life-threatening hemorrhage cannot be fully explained by endothelial cell defects alone [4, 6].

Interestingly, it has been reported that capillaries in the pituitary gland are more sensitive to VEGF inhibition than those in the brain [7]. Anti-VEGF therapy also may be a promising alternative therapy for refractory pituitary adenomas and pituitary carcinomas that are resistant to conventional treatments [8]. However, the effects of anti-VEGF therapy on the pituitary gland are not yet well understood. Pituitary apoplexy is an acute condition that causes hemorrhage and ischemic changes in the pituitary gland, and early diagnosis and treatment are important for good prognosis [9, 10]. Kasl et al. have reported a case of pituitary adenoma resulting in apoplexy after intravitreal injection of ranibizumab, a VEGF inhibitor [11]. In this case, the VEGF inhibitor was not considered to be the cause of bleeding. Pituitary apoplexy often causes hormonal dysfunction, but symptoms such as headache and fatigue are difficult to distinguish from common side effects in cancer therapy. In addition, steroids are often used in combination with cancer therapy, which may mask adrenal insufficiency.

OSI-930 is an inhibitor of receptor tyrosine kinases, including VEGF receptor 1 (VEGFR1) and 2 [12]. Clinical studies have demonstrated that OSI-930 is a well-tolerated agent which have a clinically relevant antitumor activity [13, 14]. Here, we show that OSI-930 administration induced bleeding in the pituitary gland, but not in the brain. This is the first report that anti-VEGF therapy can cause pituitary apoplexy, although it is a basic study.

## Materials and methods

### Animals

Experimental animals C57BL/6N male mice (8-12-week-old, 22-28g) were purchased from Japan SLC (Shizuoka, Japan) and housed under a 12-hour light/dark cycle (8:00–20:00 on, 20:00–8:00 off) with free access to normal autoclaved diet and water. Mouse room temperature was maintained at 23˚C. All animals used in this study were maintained and handled in accordance with the protocols approved by the Committee on Animal Research at Research Institute, Shiga Medical Center (Protocol No. R4-03). We followed NIH guidelines to treat mice and made every effort to minimize the suffering of the mice and the number of mice used for the experiments. The study was designed and carried out in compliance with the ARRIVE guidelines [15].

Male C57BL/6 mice were randomly divided into six groups with three mice, and injected intraperitoneally with 100 mg/kg body weight of OSI-930 daily for 1, 3, or 6 days. Because of additional experiments, the total number of mice in three-day and six-day OSI-930 treated groups was six and five, respectively. The dosage of OSI-930 was determined based on the study in xenograft models [12]. As control, vehicle was injected. Mice were sacrificed 24 hours after the last injection. To harvest tissues, mice were placed under 5% isoflurane until all reflexes were absent and euthanized via cardiac puncture with care to minimize pain. Subsequently, pituitary glands were subjected to H&E and immunohistochemical staining.

### Immunohistochemical analysis

Mice were perfused transcardially with 4% paraformaldehyde in PBS. Pituitary glands were isolated and embedded in Tissue-Tek O.C.T. Compound (Sakura Finetek) and cut at 30 μm thickness for immunohistochemistry. For H&E staining, the pituitary glands were fixed in

formalin for 2 days and sections were made at 10 μm thickness. Immunostaining was performed with the following antibodies as previously described: mouse anti-platelet endothelial cell adhesion molecule 1 (PECAM1) / CD31 (1:10; clone 390, 102402, BioLegend, SanDiego, USA), mouse anti-plasmalemma vesicle associated protein (PLVAP; 1:100; ab27853, Abcam, Cambridge, UK), mouse anti-platelet-derived growth factor receptor-β (PDGFR-β)/CD140b (1:100; 136002, BioLegend, SanDiego, USA), and rat anti-VE-cadherin/CD144 (1:100; clone 11D4.1, 555289, BD, NJ, USA) [16]. Alexa Fluor 568-conjugated isolectin GS-IB4 from *Griffonia simplicifolia* (1:100; I21412, Thermo Fisher Scientific, MA, USA) was used to label endothelial cells. Briefly, cryosections were incubated with primary antibodies for 24 h at 4˚, and then with secondary antibodies for 1 h at room temperature. Donkey anti-species IgG conjugated with Alexa 488 (A21206, A21208, Thermo Fisher Scientific, MA, USA) was used for a secondary antibody. Samples were then treated with DAPI. Sections were analyzed with Leica SP8 confocal laser scanning microscopy (Leica, Wetzlar, Germany).

### Vascular analysis

Sagittal sections of pituitary glands were stained using isolectin B4 conjugated with Alexa 568. To analyze vasculature, ImageJ software was used. For vessel branching analysis, capillaries around Rathke's cleft were excluded and only the anterior lobe was included. The images were processed with "tubeness" filter (sigma: 3), and then binarized images were created with near peak values of thresholds to match blood vessels. After measuring the vascular area (%), binarized images were skeletonized and the number of vascular branches and branching points per 10000 $\mu m^2$ in the anterior lobe was counted by "analyze skeleton" [17]. In the intermediate lobe, the number of vascular networks was counted. All statistical analyses were performed using EZR software [18]. The data were analyzed by Student's t-test and considered to be significant when $P < 0.05$. Results are given as means ± S.E.M.

### Quantitative real time RT-PCR analysis

Total RNA was extracted from the pituitary gland using RNA easy Mini kit (Qiagen, Hilden, Germany). Complementary DNA was obtained using a PrimeScript 1st strand cDNA synthesis kit (Takara, Shiga, Japan). Gene expressions were quantified using TB Green Premix Ex Taq II (Tli RNaseH Plus) (Takara, Shiga, Japan). Quantitative real-time PCR was performed on a LightCycler 480 system (Roche, Basel, Switzerland). *Actb* was used as a control (Mouse Housekeeping Gene Primer Set (Takara, Shiga, Japan)). The following primers were used for qPCR: *Pecam1* forward: `tcggcagacaagatgctcctggctc`, *Pecam1* reverse: `cagggtcagttgctgcccattcatcacc`, *Plvap* forward: `gctatcatcctgagcgagaagcagtgcc`, *Plvap* reverse: `tgccttctccttggccacctccatc`, *Flt1* forward: `ctgcgaagccaccgtcaacggg`, *Flt1* reverse: `gtggcggtgcagttgaggacaagag`, *Flk1* forward: `cgtaccgggacgtcgacatagcctc`, *Flk1* reverse: `ggtgatgtacacgatgccatgctggtc`, *VE-cadherin (Cdh5)* forward: `ctgccctcattgtggacaagaacaccaac`, *VE-cadherin (Cdh5)* reverse: `gacatctctggcacagatgcgttgaatacctg`.

## Results

### OSI-930 administration causes pituitary apoplexy

To evaluate adverse effects of OSI-930, we first examined the brain and pituitary gland of OSI-930-administered mice. As for the brain, no hemorrhagic change was observed in control or OSI-930 groups. In contrast, hemorrhage was frequently detected in the pituitary gland of OSI-930 groups (Fig 1). In control and groups treated with OSI-930 for one day, no bleeding

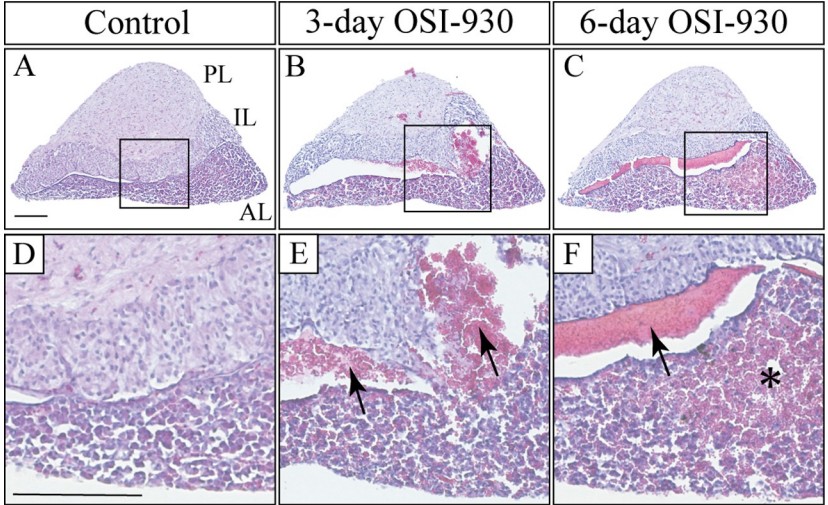

**Fig 1. OSI-930 causes pituitary apoplexy around Rathke's cleft.** H&E stained sagittal images of pituitary glands in control (A and D), three-day OSI-930-treated (B and E), and six-day OSI-930-treated (C and F) groups. Boxed regions in A-C are enlarged in D-F, respectively. In three-day OSI-930-treated group, hemorrhage occurred around Rathke's cleft (arrows; E). In six-day OSI-930-treated group, hemorrhage was exacerbated around Rathke's cleft (asterisk; F), and many small bleeds were observed throughout the anterior lobe (C and F). Bars, 200 μm. AL, anterior lobe; IL, intermediate lobe; PL, posterior lobe.

occurred in the pituitary gland (Fig 1A). Strikingly, two of three three-day OSI-930 treated mice and all of six-day OSI-930 treated mice exhibited hemorrhage in the pituitary gland (Table 1 and Fig 1B and 1C). In all cases of pituitary apoplexy, the hemorrhage occurred just around Rathke's cleft (Fig 1D–1F). In three-day OSI-930-treated group, bleeding occurred around Rathke's cleft, and accumulated within the cleft (Fig 1E). In six-day OSI-930-treated group, bleeding was more massive and also observed in wider areas of anterior and intermediate lobes (Fig 1F). These results suggest that pituitary apoplexy was observed in a treatment-day dependent manner.

## VEGFR inhibition disrupts pituitary vasculature

To investigate how the vasculature of the pituitary gland is affected by OSI-930, we stained endothelial cells with isolectin B4 (Fig 2). The capillary plexus was well developed around Rathke's cleft, a remnant space between the anterior and intermediate lobes, in control mice (Fig 2G and 2J), where pituitary stem cells has been reported to be enriched. To examine whether this isolectin B4-positive structures are blood vessels or not, we performed immuno-histochemical analysis of PECAM1 (CD31, Fig 3A–3H) and PLVAP (Fig 3I–3P). Isolectin B4 staining were observed around PECAM1-positive or PLVAP-positive blood vessels in the MCL (Fig 3A–3P), suggesting isolectin B4-positive structures are blood vessels.

**Table 1. Rates of hemorrhage in pituitary grands after OSI-930 administration.**

| | Duration of OSI-930 administration (days) | | |
|---|---|---|---|
| | 1 | 3 | 6 |
| OSI-930 | 0% (0/3) | 66.7% (4/6) | 100% (5/5) |
| Control | 0% (0/3) | 0% (0/3) | 0% (0/3) |

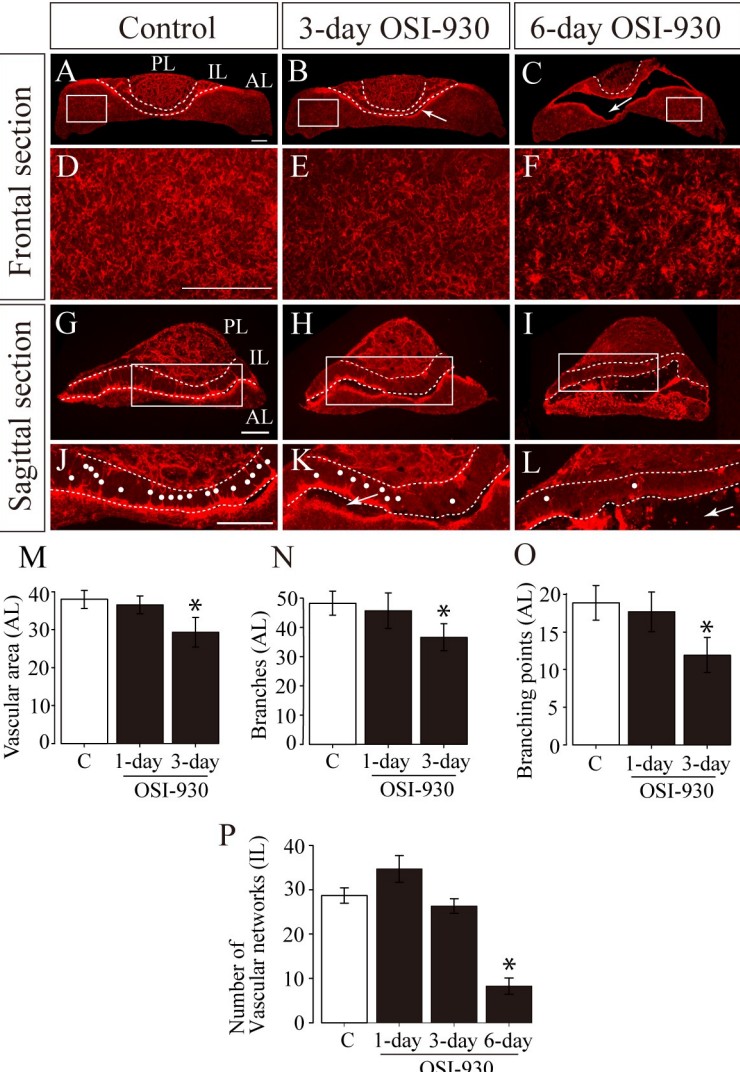

**Fig 2. VEGFR inhibition disrupts pituitary vasculature. (A-F)** Representative frontal sectional images of pituitaries labelled by isolectin B4 staining in control (A and D), three-day OSI-930-treated (B and E), and six-day OSI-930-treated (C and F) groups. Boxed regions in A-C are enlarged in D-F, respectively. **(G–L)** Representative sagittal images of pituitaries labelled by isolectin B4 staining in control (G and J), three-day OSI-930-treated (H and K), and six-day OSI-930-treated (I and L) groups. Boxed regions in G-I are enlarged in J-L, respectively. White dots indicate capillary plexus with a palisade-fashion arrangement in the intermediate lobe. In OSI-930-treated groups, hematoma cavities were observed around Rathke's cleft (arrows; B, C, K, and L), and capillary networks in the intermediate lobe were decreased compared to control (dots: J-L). In addition, capillaries around Rathke's cleft were diminished compared to control. **(M-P)** Quantitation of vascular area, branches and branching points in the anterior lobe, and vascular networks in the intermediate lobe. The vascular are (M), branches (N) and branching points (O) were decreased in the anterior lobe, and the number of capillary plexus with a palisade-fashion arrangement was decreased in the intermediate lobe (P) in a treatment-day dependent manner (* $P < 0.05$, t-test). Bars, 200 μm. AL, anterior lobe; IL, intermediate lobe; PL, posterior lobe; C, control.

After OSI-930 administration, the capillary plexus around Rathke's cleft was disrupted (Fig 2J–2L). In the anterior lobe, the vascular coverage, vascular branches and branching points were decreased in a treatment-day dependent manner (Fig 2M–2O). These OSI effects on blood vessels were also detected by the change in the expression levels of VEGFR1 and VEGFR2 but not of PECAM1, PLVAP, and VE-cadherin (Fig 4). In the intermediate lobe, the

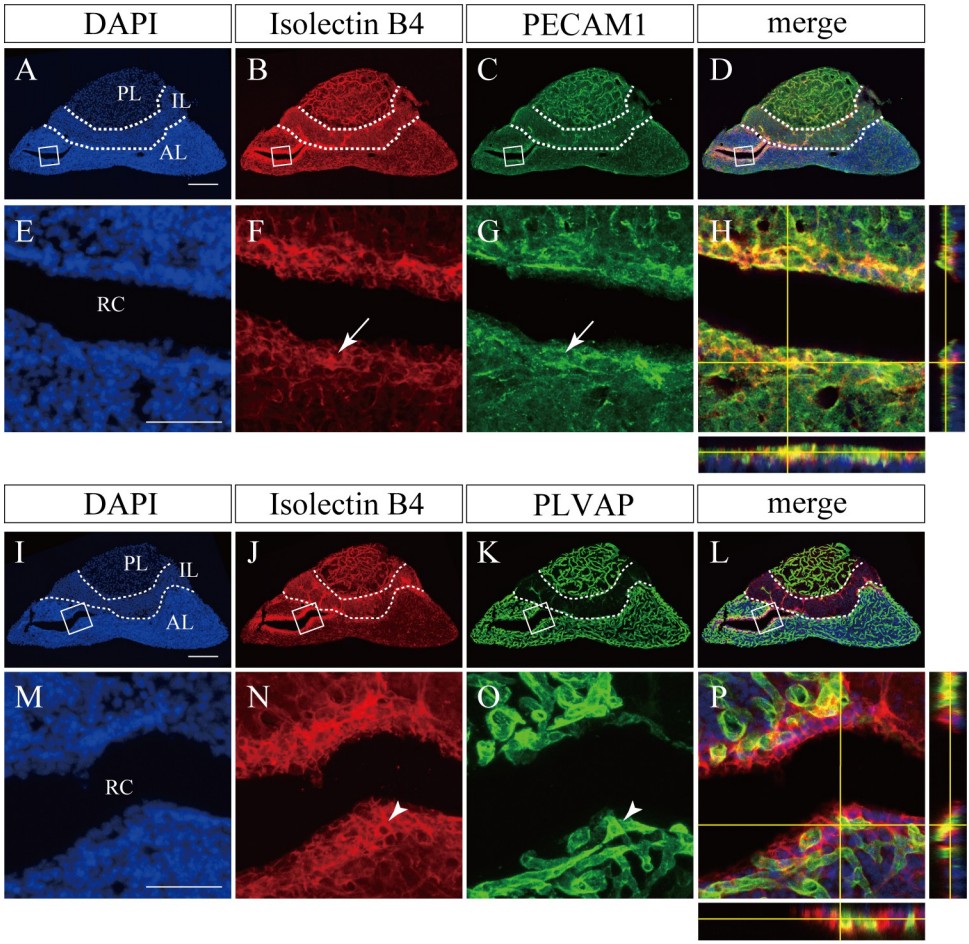

**Fig 3. Comparison of isolectin B4 staining with PECAM1 and PLVAP immunostaining.** Representative confocal immunofluorescence sagittal images with orthogonal views. Boxed regions in A-D and I-L are enlarged in E-H and M-P, respectively. Arrows and arrow heads indicate the intersection points of the XZ and YZ planes. **(A-H)** The pattern of isolectin B4 staining (red) was extremely similar to that of PECAM1 immunostaining (green), including the surrounding area of Rathke's cleft. **(I-P)** A significant number of cells positive for isolectin B4 staining were positive for PLVAP. Bars, 200 μm. AL, anterior lobe; IL, intermediate lobe; PL, posterior lobe.

number of capillary networks with a palisade-fashion arrangement was also decreased in a treatment-day dependent manner (Fig 2P). The reduction of capillary plexus around Rathke's cleft was mainly observed at the site where hemorrhage occurred (Fig 2J–2L), suggesting a causal relationship with the pathogenesis of hemorrhage.

## Contribution of pericytes and VE-cadherin to pituitary apoplexy

To analyze how the vascular structure is affected by OSI-930 in pituitary apoplexy, we examined the expression of PDGFR-β, a marker for pericytes, and VE-cadherin in pituitaries. PDGFR-β immunoreactivity were present along vessels in anterior, intermediate and posterior lobes of pituitaries of mice (Fig 5A–5H). Interestingly, PDGFR-β immunoreactivity was not present along capillaries around Rathke's cleft, where hemorrhagic changes were frequently observed after OSI-930 administration (Fig 5C and 5G). In addition, PDGFR-β expression was decreased after OSI-930 administration (Fig 5I–5P). This finding was more prominent after six-day OSI-930 treatment than after three-day treatment.

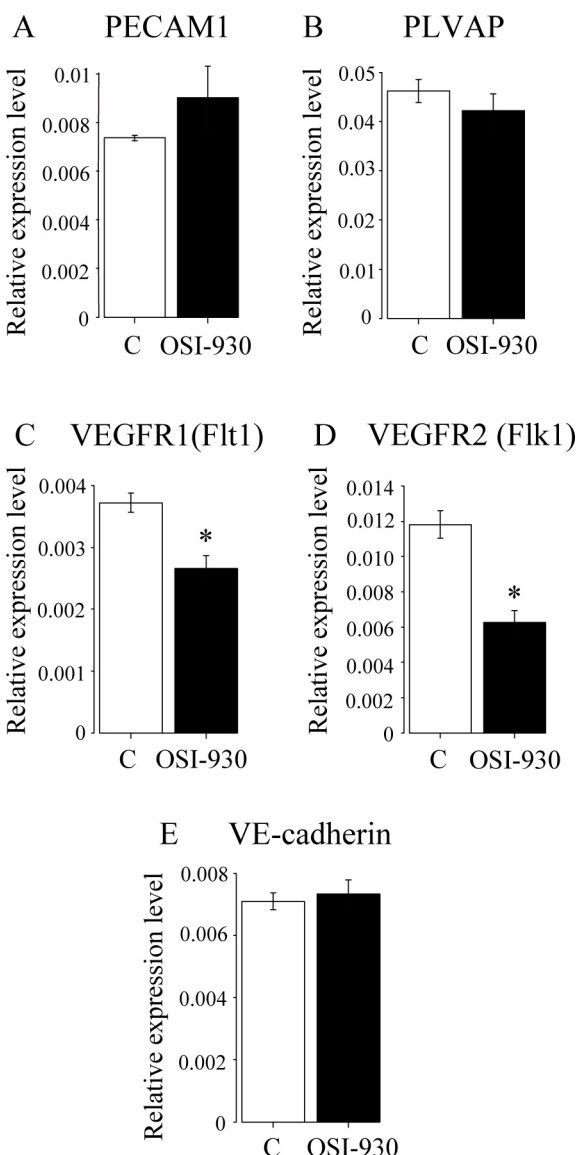

**Fig 4. Quantitative real-time RT-PCR analysis of vascular markers.** Expression levels of PECAM1 (A), PLVAP (B), VEGFR1 (C), VEGFR2 (D), and VE-cadherin (E) were analyzed. *Actb* was used as a control (* P<0.05, t-test).

VE-cadherin was expressed in vessels of anterior and posterior lobes (Fig 6A–6F). However, Few blood vessels of the intermediate lobes express VE-cadherin. As with pericytes, distribution of VE-cadherin was sparse in capillaries around Rathle's cleft, where hemorrhagic change was frequently observed after OSI-930 administration (Fig 6G–6M). Taken together, undeveloped pericytes and VE-cadherin may have influenced the susceptibility to bleeding in capillaries around Rathke's cleft.

## Discussion

In our study, OSI-930 administration induced bleeding in the pituitary gland, but not in the brain. Kamba et al. have reported that capillary regression was observed in multiple organs of mice treated with AG-013736, a small-molecule VEGFR tyrosine kinase inhibitor [7].

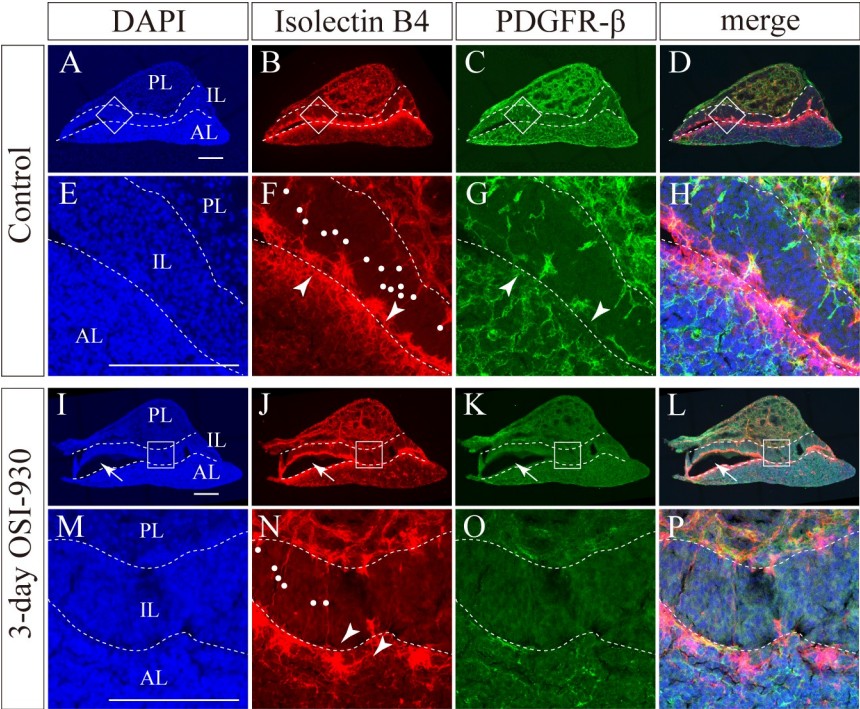

**Fig 5. Pericytes are not detected around Rathke's cleft. (A-P)** Representative sagittal images of pituitaries stained with isolectin B4 (red) and anti-PDGFR-β antibody (green). Boxed regions in A-D and I-L are enlarged in E-H and M-P, respectively. Although PDGFR-β-positive pericytes were identified along vessels throughout pituitaries, they were hardly detected around Rathke's cleft in control (arrowheads; G). In OSI-930-treated group, hematoma cavities were observed around Rathke's cleft (arrows; I-L), and capillary networks in the intermediate lobe were decreased compared to control (dots: F and N). In addition, capillaries around Rathke's cleft were diminished compared to control (arrowheads; F and N). In OSI-930-teated group, pericytes were decreased compared to control (C, G, K and O). Bars, 200 μm. AL, anterior lobe; IL, intermediate lobe; PL, posterior lobe.

Capillary regression was found more frequently in anterior (24%) and posterior (41%) pituitaries than in brains (5%). However, details of vascular changes were not examined in pituitaries. These data may explain the reason why OSI-930 induced bleeding in the pituitary, but not in the brain.

VEGFR-2 (Flk-1) was expressed in endothelial cells in rat and human pituitary glands [19, 20]. Furube E et al. have reported that administration of AZD2171, a receptor tyrosine kinases inhibitor for VEGFRs, decreased vascular density and the proliferation of endothelial cells in posterior pituitaries of mice [21]. However, pituitary apoplexy was not observed in their study. Only posterior lobes, but not anterior or intermediate lobes were examined. In contrast, we examined vascular changes in both anterior and intermediate lobes after administration of OSI-930. Capillary regression was more prominent in the vascular plexus around Rathke's cleft, which might be the reason why pituitary apoplexy often occurs around Rathke's cleft after OSI-930 treatment. Since MCLs and folliculo-stellate cells are also positive for isolectin B4 in the pituitary gland [22], we compared the pattern of isolectin B4 staining with those of CD31 and PLVAP immunostaining. The pattern of CD31 immunostaining was extremely similar to that of isolectin B4 staining, including the MCL portion. Isolectin B4 staining has the limitation that non-endothelial cells may also be positive. However, in the present study, this staining method was considered feasible for the purpose of analyzing pituitary vessels.

In the human pituitary gland, VEGFR1 and VEGFR2 were expressed in endocrine cells and vascular structures, respectively [20]. In the rat pituitary gland, VEGFR1 and VEGFR2 were

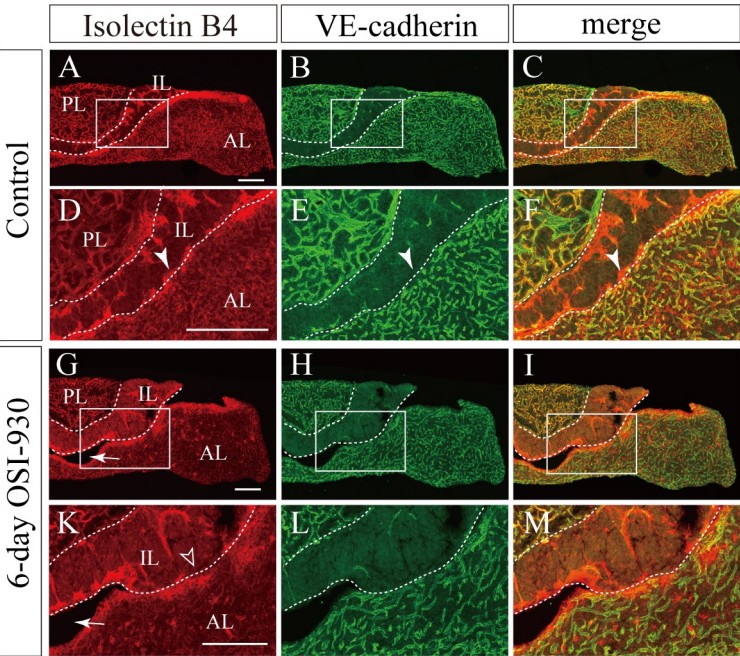

**Fig 6. Few blood vessels of the intermediate lobes express VE-cadherin. (A-M)** Representative frontal sectional images of pituitaries stained with isolectin B4 (red) and anti-VE-cadherin antibody (green). Boxed regions in A-C and G-I are enlarged in D-F and K-M, respectively. Although VE-cadherin was identified in both anterior and posterior pituitaries, it was hardly detected in the intermediate lobe and around Rathke's cleft in control (arrowheads; D-F). In OSI-930-treated group, hematoma cavities were observed around Rathke's cleft (arrows; G and K). In addition, capillaries around Rathke's cleft were diminished compared to control (open arrowhead; K). Bars, 200 μm. AL, anterior lobe; IL, intermediate lobe; PL, posterior lobe.

also expressed in endocrine and endothelial cells [19, 20]. In our study, the expression of VEGFR1 was decreased after OSI-930 administration. This may reflect hormonal cell destruction by pituitary apoplexy. The expression of VEGFR2 was also decreased. This may reflect the vascular regression in pituitaries.

Pericytes cover the microvascular wall, and contribute to the mechanical stability of the capillary wall [23]. PDGF-B or PDGFR-β deficient mouse embryos lack microvascular pericytes and develop numerous microvascular microaneurysms resulting in hemorrhage [23, 24]. VE-cadherin has a role in the maintenance of cell-cell junction stabilization and regulation of vascular barrier integrity [25, 26]. In our study, hemorrhage was frequently observed in the vascular plexus of intermediate lobes around Rathke's cleft, where pericytes did not exist and few vessels express VE-cadherin. The VE-cadherin-negative capillaries without pericytes around Rathke's cleft might be susceptible for bleeding. The capillaries distributed in the pituitary gland are characteristically rich in fenestrations [27, 28]. It has been reported that VEGFR inhibitor administration reduced endothelial fenestrations in thyroid perifollicular capillaries, renal glomerular capillaries, and capillaries in islet-cell tumors of RIP-Tag2 transgenic mice [7, 29]. To date, no studies have been reported on fenestration changes after VEGFR inhibitor administration in pituitary capillaries. Due to lack of access to an electron microscopy, we could not examine ultrastructural changes in pituitary capillary endothelial fenestrations. Pituitary apoplexy may be caused due to changes of endothelial fenestration with OSI-930 treatment.

Our study has a limitation that should be noted. We performed experiments with higher doses than in clinical trials in order to more clearly evaluate the adverse effects of OSI-930. For

this reason, it is likely that the incidence of pituitary apoplexy induced by anti-VEGF therapy will be lower in actual clinical practice. However, we consider that our study is meaningful, because it raises awareness of adverse effects on the pituitary gland during clinical application of anti-VEGF agents.

## Conclusion

Our study demonstrates the possibility that anti-VEGF agents have a risk of pituitary apoplexy. Regarding adverse effects of anti-VEGF therapy for cancer, no case of pituitary apoplexy has been previously reported in either basic or clinical studies. This is the first report that anti-VEGF therapy can cause pituitary apoplexy, although it is a basic study. Pituitary apoplexy should be kept in mind as an adverse effect of anti-VEGF therapy. If patients treated with anti-VEGF agents complain of symptoms such as headache and fatigue, brain imaging and hormonal examinations should be considered.

## Supporting information

**S1 File. Raw data of tissue sections used for vascular analysis.**
(PPTX)

## Author Contributions

**Conceptualization:** Kenji Tanigaki, Masato Hojo.

**Data curation:** Yoshito Sugita, Shigeki Takada, Kenji Tanigaki.

**Formal analysis:** Kenji Tanigaki, Masato Hojo.

**Investigation:** Yoshito Sugita, Shigeki Takada, Kenji Tanigaki, Kazue Muraki, Munehiro Uemura.

**Supervision:** Susumu Miyamoto.

**Writing – original draft:** Yoshito Sugita.

**Writing – review & editing:** Kenji Tanigaki, Masato Hojo.

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
