## [Decision Letter · Decision Letter 0]

18 Jul 2022

PONE-D-22-14999Inhibition of VEGF receptors induces pituitary apoplexy: an experimental study in micePLOS ONE

Dear Dr. Hojo,

Thank you for submitting your manuscript to PLOS ONE. After careful consideration, we feel that it has merit but does not fully meet PLOS ONE’s publication criteria as it currently stands. Therefore, we invite you to submit a revised version of the manuscript that addresses the points raised during the review process.

We look forward to receiving your revised manuscript.

Kind regards,

Takahiro Nemoto, Ph.D

Academic Editor

PLOS ONE

Journal Requirements:

Additional Editor Comments:

We apologize for the time it took. Two reviewers have commented, so I'd like to ask for improvements to make the paper better.

Reviewers' comments:

Reviewer's Responses to Questions

**Comments to the Author**

1. Is the manuscript technically sound, and do the data support the conclusions?

Reviewer #1: Yes

Reviewer #2: No

2. Has the statistical analysis been performed appropriately and rigorously? 

Reviewer #1: Yes

Reviewer #2: Yes

3. Have the authors made all data underlying the findings in their manuscript fully available?

Reviewer #1: Yes

Reviewer #2: Yes

4. Is the manuscript presented in an intelligible fashion and written in standard English?

Reviewer #1: Yes

Reviewer #2: Yes

5. Review Comments to the Author

Reviewer #1: To authors

It is a very intriguing issue whether inhibition of VEGF receptor is involved in the pituitary apoplexy. In this study, the authors provide that the mice administered with OSI-930, an inhibitor of receptor tyrosine kinases including VEGF receptor 1 and 2, exhibited hemorrhage in the pituitary gland. Although this study contains some new findings, this reviewer considers that more extensive study is required for the anti-vascular endothelial growth factor (VEGF) therapy.

Major comments:

Authors used Alexa Fluor 568-conjugated isolectin GS-IB4 from Griffonia simplicifolia (1:100; Thermo Fisher Scientific) for labeling endothelial cells in the anterior lobe of mouse pituitary. It is not well known, but I do that GS-IB4 react the folliculo-stellated cells and marginal cells in the intermediate lobe in mouse anterior pituitary. GS-IB4 is adequacy for detecting endothelial cells in mouse pituitary. As shown in Fig. 2, 3, and 4, GS-IB4-posotive cells located in the marginal cell layer (MCL) that face Rathke’s cleft in the anterior lobe and intermediate lobe form the primary stem cell niche in the adult adenohypophysis. Especially, there are few capillaries in the MCL of anterior lobe side. Authors should fundamentally change.

Minor comments:

Authors used sections at 30 μm thickness for H&E staining? It is not fine data (Fig. 1A-F). Authors should use 6-10 μm thickness sections.

Page 2 line 26: Two of three three-day?

Page 2 line 31: The expression of anti-?

Page 6 line 90: Lica?

Reviewer #2: This study is a morphological analysis of the effects of the clinically used VEGF-A receptor inhibitor OSI-930 on the pituitary when administered to mice. The data that administration induces bleeding is morphologically shown, and I think it is a very interesting paper. However, there are many inadequate points in the content, so please refer to the comments below and make corrections.

Major comments.

1. Experience has shown that isolectin B4 is positive in non-endothelial cells that make up the marginal cell layer (MCL) around Rathke ’s cleft. In this study, authors thought that all isolectin B4-positive cells are endothelial cells, but special attention should be paid to the cells around MCL. Since PECAM1 and PLVAP/PV-1 are common markers for pituitary endothelial cells, so you should also use that antibody to confirm that the isolectin B4-positive cells around the MCL are endothelial cells.

2. The intermediate lobe of the pituitary in rodents such as mouse and rat is composed of a lobule structure, which is a tissue with few blood vessels with only a few capillaries between the lobule structures. On the other hand, the anterior lobe adjacent to the intermediate lobe is rich in fenestrated capillaries that connect from the hypophyseal portal system, but this paper rarely observes morphological changes in the capillaries of the anterior lobe. The vasculature of the anterior lobe is the most important structure for pituitary function and should be analyzed as well. Observation of the anterior lobe structure is easy to observe in the frontal section but not sagittal section, you should analyze changes in the vascular structure of the anterior lobe.

3. VE-cadherin is essential for the adhesion of endothelial cells, and its reduced expression usually leads to disruption of vascular structure. In Fig.4, the localization of VE-cadherin is investigated by immunohistochemistry, but it is not clear from this data alone whether the vascular structure is broken or the expression of VE-cadherin is only reduced. You should clarify the change in the expression level of marker genes (Pecam1, Plvap, Flt1, Flk1, etc.) including VE-cadherin (Cdh5) in the pituitary by qPCR.

4. The capillaries distributed in the pituitary are fenestrated type. Because the vasculature was confirmed in the pituitary of OSI-930-treated mice by isolectin B4 staining, pituitary apoplexy may be due to changes of endothelial fenestration with OSI-930 treatment. It may be necessary to observe the ultrafine structure of endothelial cells using an electron microscopy to reveal the fenestra structure in the pituitary of mouse with OSI-930 treatment.

Mainer comments.

1． Separate the figure legend from the text and summarize it.

2． Please correct line 90 Lica to Lieca.

3． Please correct line 229 VRGF to VEGF.

4． The notation of AL and PL in Figure 1A is reversed.

6. PLOS authors have the option to publish the peer review history of their article (what does this mean?). If published, this will include your full peer review and any attached files.

Reviewer #1: No

Reviewer #2: No

---

## [Author Response · Author response to Decision Letter 0]

3 Oct 2022

Comments for Journal Requirements (2):

1. Please provide details on methods of sacrifice.

Response: We have revised the Animals section in Materials and method as follows.

“ To harvest tissues, mice were placed under 5% isoflurane until all reflexes were absent and euthanized via cardiac puncture with care to minimize pain. “ (L88)

“Experimental animals C57BL/6N male mice (8-12-week-old, 22-28g) were purchased from Japan SLC (Shizuoka, Japan) and housed under a 12-hour light/dark cycle (8:00–20:00 on, 20:00–8:00 off) with free access to normal autoclaved diet and water. Mouse room temperature was maintained at 23 ºC.” (L74)

“We followed NIH guidelines to treat mice and made every effort to minimize the suffering of the mice and the number of mice used for the experiments. The study was designed and carried out in compliance with the ARRIVE guidelines [15]. Male C57BL/6 mice were randomly divided into six groups with three mice, and injected intraperitoneally with 100 mg/kg body weight of OSI-930 daily for 1, 3, or 6 days. Because of additional experiments, the total number of mice in three-day and six-day OSI-930 treated groups was six and five, respectively. The dosage of OSI-930 was determined based on the study in xenograft models” (L79)

Comments for Journal Requirements (1):

1. Please ensure that your manuscript meets PLOS ONE's style requirement.

Response: We have reviewed our manuscript to ensure it meet PLOS ONE's style requirement.

2. Please provide additional information regarding the experiments involving animals.

Response: We have revised the Animals section in Materials and method as follows.

“ To harvest tissues, mice were placed under 5% isoflurane until all reflexes were absent and euthanized via cardiac puncture with care to minimize pain. “ (L88)

“Experimental animals C57BL/6N male mice (8-12-week-old, 22-28g) were purchased from Japan SLC (Shizuoka, Japan) and housed under a 12-hour light/dark cycle (8:00–20:00 on, 20:00–8:00 off) with free access to normal autoclaved diet and water. Mouse room temperature was maintained at 23 ºC.” (L74)

“We followed NIH guidelines to treat mice and made every effort to minimize the suffering of the mice and the number of mice used for the experiments. The study was designed and carried out in compliance with the ARRIVE guidelines [15]. Male C57BL/6 mice were randomly divided into six groups with three mice, and injected intraperitoneally with 100 mg/kg body weight of OSI-930 daily for 1, 3, or 6 days. Because of additional experiments, the total number of mice in three-day and six-day OSI-930 treated groups was six and five, respectively. The dosage of OSI-930 was determined based on the study in xenograft models” (L79)

3. All PLOS journals require that the minimal data set be made fully available.

Response: We have uploaded the raw data for the analysis as supporting information file.

4. PLOS does not permit references to inaccessible data.

Response: We have removed from Results section the results that do not display any data. All results now properly display the data on which they are based.

Comments for Reviewers:

Reviewer: 1

Major comments:

“It is not well known, but I do that GS-IB4 react the folliculo-stellated cells and marginal cells in the intermediate lobe in mouse anterior pituitary.”

Response: We have newly harvested the pituitary gland from mice, and performed double staining with ILB4 and anti-PECAM1 antibody, and double staining with ILB4 and anti-PLVAP antibody. The results suggest that ILB4 staining is positive for structures other than vessels in the MCL, but can also be used to some extent in order to analyze pituitary vessels, although there are some limitations. We have added Fig 3, and added sentences as follows.

“To examine whether this isolectin B4-positive structures are blood vessels or not, we performed immunohistochemical analysis of PECAM1 (CD31, Figs 3A-H) and PLVAP (Figs 3I-P). Isolectin B4 staining were observed around PECAM1-positive or PLVAP-positive blood vessels in the MCL (Figs 3A-P), suggesting isolectin B4-positive structures are blood vessels.” (L168)

“Fig 3. Comparison of isolectin B4 staining with PECAM1 and PLVAP immunostaining. Representative confocal immunofluorescence sagittal images with orthogonal views. Boxed regions in A-D and I-L are enlarged in E-H and M-P, respectively. Arrows and arrow heads indicate the intersection points of the XZ and YZ planes. (A-H) The pattern of isolectin B4 staining (red) was extremely similar to that of PECAM1 immunostaining (green), including the surrounding area of Rathke's cleft. (I-P) A significant number of cells positive for isolectin B4 staining were positive for PLVAP. Bars, 200 µm. AL, anterior lobe; IL, intermediate lobe; PL, posterior lobe.” (L192)

Minor comments:

“Authors used sections at 30 μm thickness for H&E staining? It is not fine data (Fig. 1A-F). Authors should use 6-10 μm thickness sections.”

Response: We have newly harvested the pituitary gland from mice, made sections at 10 μm thickness, and performed H&E staining. We have revised Fig 1, and added sentences as follows.

“For H&E staining, the pituitary glands were fixed in formalin for 2 days and sections were made at 10 μm thickness.” (L96)

“Page 2 line 26: Two of three three-day?”

Response: We have revised the sentence as follows. The number of mice analyzed has increased with additional experiments.

“Four of six mice treated for three days and all of five mice treated for six days exhibited hemorrhage in the pituitary gland.” (L26)

“Page 2 line 31: The expression of anti-?”

Response: We have corrected the sentence as follows.

“Few blood vessels around Rathke’s cleft of the intermediate lobe express VE-cadherin and are covered with platelet-derived growth factor receptor-β (PDGFR-β)-positive cells” (L31)

“Page 6 line 90: Lica?”

Response: We have corrected the sentence as follows.

“confocal laser scanning microscopy (Leica).” (L108)

Reviewer: 2

Major comments.

1. Since PECAM1 and PLVAP/PV-1 are common markers for pituitary endothelial cells, so you should also use that antibody to confirm that the isolectin B4-positive cells around the MCL are endothelial cells.

Response: We have newly harvested the pituitary gland from mice, and performed double staining with ILB4 and anti-PECAM1 antibody, and double staining with ILB4 and anti-PLVAP antibody. The results suggest that ILB4 staining is positive for structures other than vessels in the MCL, but can also be used to some extent in order to analyze pituitary vessels, although there are some limitations. We have added Fig 3, and added sentences as follows.

“To examine whether this isolectin B4-positive structures are blood vessels or not, we performed immunohistochemical analysis of PECAM1 (CD31, Figs 3A-H) and PLVAP (Figs 3I-P). Isolectin B4 staining were observed around PECAM1-positive or PLVAP-positive blood vessels in the MCL (Figs 3A-P), suggesting isolectin B4-positive structures are blood vessels.” (L168)

“Fig 3. Comparison of isolectin B4 staining with PECAM1 and PLVAP immunostaining. Representative confocal immunofluorescence sagittal images with orthogonal views. Boxed regions in A-D and I-L are enlarged in E-H and M-P, respectively. Arrows and arrow heads indicate the intersection points of the XZ and YZ planes. (A-H) The pattern of isolectin B4 staining (red) was extremely similar to that of PECAM1 immunostaining (green), including the surrounding area of Rathke's cleft. (I-P) A significant number of cells positive for isolectin B4 staining were positive for PLVAP. Bars, 200 µm. AL, anterior lobe; IL, intermediate lobe; PL, posterior lobe.” (L192)

2. Observation of the anterior lobe structure is easy to observe in the frontal section but not sagittal section, you should analyze changes in the vascular structure of the anterior lobe.

Response: Although there is a limit to the number of mice that can be prepared for additional experiments, we administered new doses of OSI to as many mice as possible, harvested the pituitary gland, and, whenever possible, created a new coronal section and analyzed the anterior lobe. We have revised Fig 2 and 6, and revised sentences as follows.

“Fig 2. VEGFR inhibition disrupts pituitary vasculature. (A-F) Representative frontal sectional images of pituitaries labelled by isolectin B4 staining in control (A and D), three-day OSI-930-treated (B and E), and six-day OSI-930-treated (C and F) groups. Boxed regions in A-C are enlarged in D-F, respectively. (G–L) Representative sagittal images of pituitaries labelled by isolectin B4 staining in control (G and J), three-day OSI-930-treated (H and K), and six-day OSI-930-treated (I and L) groups. Boxed regions in G-I are enlarged in J-L, respectively. White dots indicate capillary plexus with a palisade-fashion arrangement in the intermediate lobe. In OSI-930-treated groups, hematoma cavities were observed around Rathke’s cleft (arrows; B, C, K, and L), and capillary networks in the intermediate lobe were decreased compared to control (dots: J-L). In addition, capillaries around Rathke’s cleft were diminished compared to control. (M-P) Quantitation of vascular area, branches and branching points in the anterior lobe, and vascular networks in the intermediate lobe. The vascular are (M), branches (N) and branching points (O) were decreased in the anterior lobe, and the number of capillary plexus with a palisade-fashion arrangement was decreased in the intermediate lobe (P) in a treatment-day dependent manner (* P < 0.05, t-test). Bars, 200 µm. AL, anterior lobe; IL, intermediate lobe; PL, posterior lobe; C, control.” (L174)

“In the anterior lobe, the vascular coverage, vascular branches and branching points were decreased in a treatment-day dependent manner (Figs 2M-O).” (L202)

“VE-cadherin was expressed in vessels of anterior and posterior lobes (Figs 6A-F). However, Few blood vessels of the intermediate lobes express VE-cadherin. As with pericytes, distribution of VE-cadherin was sparse in capillaries around Rathle’s cleft, where hemorrhagic change was frequently observed after OSI-930 administration (Figs 6G-M). Taken together, undeveloped pericytes and VE-cadherin may have influenced the susceptibility to bleeding in capillaries around Rathke’s cleft. Fig 6. Few blood vessels of the intermediate lobes express VE-cadherin. (A-M) Representative frontal sectional images of pituitaries stained with isolectin B4 (red) and anti-VE-cadherin antibody (green).” (L240)

3. You should clarify the change in the expression level of marker genes (Pecam1, Plvap, Flt1, Flk1, etc.) including VE-cadherin (Cdh5) in the pituitary by qPCR.

Response: We have newly analyzed vascular markers including Pecam1, Plvap, Flt1, Flk1, and VE-cadherin (Cdh5) in the pituitary gland by qRT-PCR. We have added Fig 4 and added sentences as follows.

“Quantitative real time RT-PCR analysis

Total RNA was extracted from the pituitary gland using RNA easy Mini kit (Qiagen, Hilden, Germany). Complementary DNA was obtained using a PrimeScript 1st strand cDNA synthesis kit (Takara, Shiga, Japan). Gene expressions were quantified using TB Green　Premix Ex Taq II　(Tli RNaseH Plus) (Takara, Shiga, Japan). Quantitative real-time PCR was performed on a LightCycler 480 system (Roche, Basel, Switzerland). Actb was used as a control (Mouse Housekeeping Gene Primer Set (Takara, Shiga, Japan)). The following primers were used for qPCR: Pecam1 forward: tcggcagacaagatgctcctggctc, Pecam1 reverse: cagggtcagttgctgcccattcatcacc, Plvap forward: gctatcatcctgagcgagaagcagtgcc, Plvap reverse: tgccttctccttggccacctccatc, Flt1 forward: ctgcgaagccaccgtcaacggg, Flt1 reverse: gtggcggtgcagttgaggacaagag, Flk1 forward: cgtaccgggacgtcgacatagcctc, Flk1 reverse: ggtgatgtacacgatgccatgctggtc, VE-cadherin (Cdh5) forward: ctgccctcattgtggacaagaacaccaac, VE-cadherin (Cdh5) reverse: gacatctctggcacagatgcgttgaatacctg.” (L123)

“These OSI effects on blood vessels were also detected by the change in the expression levels of VEGFR1 and VEGFR2 but not of PECAM1, PLVAP, and VE-cadherin (Fig 4).” (L203)

“Fig 4. Quantitative real-time RT-PCR analysis of vascular markers. Expression levels of PECAM1 (A), PLVAP (B), VEGFR1 (C), VEGFR2 (D), and VE-cadherin (E) were analyzed. Actb was used as a control (* P<0.05, t-test).” (L212)

“In our study, the expression of VEGFR1 was decreased after OSI-930 administration. This may reflect hormonal cell destruction by pituitary apoplexy. The expression of VEGFR2 was also decreased. This may reflect the vascular regression in pituitaries.“ (L283)

4. Pituitary apoplexy may be due to changes of endothelial fenestration with OSI-930 treatment. It may be necessary to observe the ultrafine structure of endothelial cells using an electron microscopy to reveal the fenestra structure in the pituitary of mouse with OSI-930 treatment.

Response: An electron microscopic analysis was attempted, but technically could not be performed. Therefore, the text has been revised as follows to clarify the limitations of this paper. We have added sentences as follows.

“The capillaries distributed in the pituitary gland are characteristically rich in fenestrations [26, 27].　It has been reported that VEGFR inhibitor administration reduced endothelial fenestrations in thyroid perifollicular capillaries, renal glomerular capillaries, and capillaries in islet-cell tumors of RIP-Tag2 transgenic mice [7, 28]. To date, no studies have been reported on fenestration changes after VEGFR inhibitor administration in pituitary capillaries. Due to lack of access to an electron microscopy, we could not examine ultrastructural changes in pituitary capillary endothelial fenestrations. Pituitary apoplexy may be caused due to changes of endothelial fenestration with OSI-930 treatment.” (L295)

Minor comments.

1． Separate the figure legend from the text and summarize it.

Response: Sorry, but PLOS ONE submissions requirements say that Each figure caption should appear directly after the paragraph in which they are first cited.

2． Please correct line 90 Lica to Leica.

Response: We have corrected it. (L108)

3． Please correct line 229 VRGF to VEGF.

Response: We have corrected it. (L309)

4． The notation of AL and PL in Figure 1A is reversed.

Response: We have corrected it. (Fig 1)

---

## [Decision Letter · Decision Letter 1]

14 Nov 2022

PONE-D-22-14999R1Inhibition of VEGF receptors induces pituitary apoplexy: an experimental study in micePLOS ONE

Dear Dr. Hojo,

Thank you for submitting your manuscript to PLOS ONE. After careful consideration, we feel that it has merit but does not fully meet PLOS ONE’s publication criteria as it currently stands. Therefore, we invite you to submit a revised version of the manuscript that addresses the points raised during the review process.

We look forward to receiving your revised manuscript.

Kind regards,

Takahiro Nemoto, Ph.D

Academic Editor

PLOS ONE

Journal Requirements:

Reviewers' comments:

Reviewer's Responses to Questions

**Comments to the Author**

1. If the authors have adequately addressed your comments raised in a previous round of review and you feel that this manuscript is now acceptable for publication, you may indicate that here to bypass the “Comments to the Author” section, enter your conflict of interest statement in the “Confidential to Editor” section, and submit your "Accept" recommendation.

Reviewer #1: (No Response)

Reviewer #2: (No Response)

2. Is the manuscript technically sound, and do the data support the conclusions?

Reviewer #1: Yes

Reviewer #2: Yes

3. Has the statistical analysis been performed appropriately and rigorously? 

Reviewer #1: Yes

Reviewer #2: Yes

4. Have the authors made all data underlying the findings in their manuscript fully available?

Reviewer #1: Yes

Reviewer #2: Yes

5. Is the manuscript presented in an intelligible fashion and written in standard English?

Reviewer #1: Yes

Reviewer #2: Yes

6. Review Comments to the Author

Reviewer #1: (No Response)

Reviewer #2: I think that the revised version has been well corrected with reference to my points. However, there are some points that need to be fixed. Please refer to the following and correct it.

1. In materials and methods, you must indicate the city and country of vendors of the reagents and equipment. Also, please add the product numbers of all antibodies.

2. In references, please delete the "conflict of interest" part after line 382 because it is unnecessary.

7. PLOS authors have the option to publish the peer review history of their article (what does this mean?). If published, this will include your full peer review and any attached files.

Reviewer #1: No

Reviewer #2: No

---

## [Author Response · Author response to Decision Letter 1]

17 Nov 2022

Comments for Journal Requirements:

Response: Three authors (MH, KT, and YS) independently and carefully checked the reference list to ensure that retracted papers were not cited. Our manuscript did not cite retracted papers. Because the description of reference 15 was inadequate, we have corrected it as follows.

“15. Percie du Sert N, Hurst V, Ahluwalia A, Alam S, Avey MT, Baker M, et al. The ARRIVE guidelines 2.0: Updated guidelines for reporting animal research. PLoS Biol. 2020;18(7):e3000410. Epub 2020/07/15. doi: 10.1371/journal.pbio.3000410. PubMed PMID: 32663219; PubMed Central PMCID: PMCPMC7360023.“ (L380)

Comments for Reviewers:

Reviewer #2:

1. In materials and methods, you must indicate the city and country of vendors of the reagents and equipment. Also, please add the product numbers of all antibodies.

Response: We have revised the materials and methods section as follows.

“mouse anti-platelet endothelial cell adhesion molecule 1 (PECAM1) / CD31 (1:10; clone 390, 102402, BioLegend, SanDiego, USA), mouse anti-plasmalemma vesicle associated protein (PLVAP; 1:100; ab27853, Abcam, Cambridge, UK), mouse anti-platelet-derived growth factor receptor-β (PDGFR-β)/CD140b (1:100; 136002, BioLegend, SanDiego, USA), and rat anti-VE-cadherin/CD144 (1:100; clone 11D4.1, 555289, BD, NJ, USA) [16]. Alexa Fluor 568-conjugated isolectin GS-IB4 from Griffonia simplicifolia (1:100; I21412, Thermo Fisher Scientific, MA, USA) was used to label endothelial cells. Briefly, cryosections were incubated with primary antibodies for 24 h at 4 ℃, and then with secondary antibodies for 1 h at room temperature. Donkey anti-species IgG conjugated with Alexa 488 (A21206, A21208, Thermo Fisher Scientific, MA, USA) was used for a secondary antibody. Samples were then treated with DAPI. Sections were analyzed with Leica SP8 confocal laser scanning microscopy (Leica, Wetzlar, Germany). “ (L99)

2. In references, please delete the "conflict of interest" part after line 382 because it is unnecessary.

Response: We have deleted the "conflict of interest" part in reference 15 as follows.

“15. Percie du Sert N, Hurst V, Ahluwalia A, Alam S, Avey MT, Baker M, et al. The ARRIVE guidelines 2.0: Updated guidelines for reporting animal research. PLoS Biol. 2020;18(7):e3000410. Epub 2020/07/15. doi: 10.1371/journal.pbio.3000410. PubMed PMID: 32663219; PubMed Central PMCID: PMCPMC7360023.“ (L380)

---

## [Editor Report · Decision Letter 2]

23 Nov 2022

PONE-D-22-14999R2Inhibition of VEGF receptors induces pituitary apoplexy: an experimental study in micePLOS ONE

Dear Dr. Hojo,

Thank you for submitting your manuscript to PLOS ONE. After careful consideration, we feel that it has merit but does not fully meet PLOS ONE’s publication criteria as it currently stands. Therefore, we invite you to submit a revised version of the manuscript that addresses the points raised during the review process.

This review has been carefully reviewed and revised by two reviewers. However, some modifications are required. It is necessary to revise the manuscript according to the reviewer's comments or to respond to the comments.

We look forward to receiving your revised manuscript.

Kind regards,

Takahiro Nemoto, Ph.D

Academic Editor

PLOS ONE
---

## [Author Response · Author response to Decision Letter 2]

24 Nov 2022

Comments for Journal Requirements:

Journal Requirements:

“Please review your reference list to ensure that it is complete and correct. If you have cited papers that have been retracted, please include the rationale for doing so in the manuscript text, or remove these references and replace them with relevant current references. Any changes to the reference list should be mentioned in the rebuttal letter that accompanies your revised manuscript. If you need to cite a retracted article, indicate the article’s retracted status in the References list and also include a citation and full reference for the retraction notice. “

Response: We again checked the reference list to ensure that retracted papers were not cited. Our manuscript did not cite retracted papers. In addition, we installed the latest version of the style file in EndNote and reformatted the reference list. We believe that this work has brought the reference list into full compliance with the submission rules.

---

## [Editor Report · Decision Letter 3]

29 Nov 2022

PONE-D-22-14999R3Inhibition of VEGF receptors induces pituitary apoplexy: an experimental study in micePLOS ONE

Dear Dr. Hojo,

Thank you for submitting your manuscript to PLOS ONE. After careful consideration, we feel that it has merit but does not fully meet PLOS ONE’s publication criteria as it currently stands. Therefore, we invite you to submit a revised version of the manuscript that addresses the points raised during the review process.

The revised manuscript was carefully reviewed again by two reviewers.  Although the revised manuscript was an improvement, some concerns still remain.  The editor wish you  to resubmit the revised manuscript.

We look forward to receiving your revised manuscript.

Kind regards,

Takahiro Nemoto, Ph.D

Academic Editor

PLOS ONE

Journal Requirements:

Additional Editor Comments:

This revised manuscript has been reviewed by 2 peer reviewers. As a result of the revision, this paper has been improved, but some concerns still remain. Editors expect authors to resubmit in response to reviewers' requests.
---

## [Author Response · Author response to Decision Letter 3]

4 Dec 2022

Comments for Journal Requirements:

Journal Requirements:

“Please review your reference list to ensure that it is complete and correct. If you have cited papers that have been retracted, please include the rationale for doing so in the manuscript text, or remove these references and replace them with relevant current references. Any changes to the reference list should be mentioned in the rebuttal letter that accompanies your revised manuscript. If you need to cite a retracted article, indicate the article’s retracted status in the References list and also include a citation and full reference for the retraction notice. “

Response: The link between DOI numbers and articles was inadequate. Therefore, we have corrected this problem, and proper linking is now possible. In addition, we have again checked the reference list to ensure that retracted papers were not cited. Our manuscript did not cite retracted papers.

---

## [Decision Letter · Decision Letter 4]

12 Dec 2022

Inhibition of VEGF receptors induces pituitary apoplexy: an experimental study in mice

PONE-D-22-14999R4

Dear Dr. Hojo,

We’re pleased to inform you that your manuscript has been judged scientifically suitable for publication and will be formally accepted for publication once it meets all outstanding technical requirements.

Kind regards,

Takahiro Nemoto, Ph.D

Academic Editor

PLOS ONE

Additional Editor Comments (optional):

This manuscript was reviewed by two reviewers. As a result of multiple rounds of peer review, the manuscript was accepted for publication by two reviewers.

Reviewers' comments:

Reviewer's Responses to Questions

**Comments to the Author**

1. If the authors have adequately addressed your comments raised in a previous round of review and you feel that this manuscript is now acceptable for publication, you may indicate that here to bypass the “Comments to the Author” section, enter your conflict of interest statement in the “Confidential to Editor” section, and submit your "Accept" recommendation.

Reviewer #2: All comments have been addressed

2. Is the manuscript technically sound, and do the data support the conclusions?

Reviewer #2: Yes

3. Has the statistical analysis been performed appropriately and rigorously? 

Reviewer #2: Yes

4. Have the authors made all data underlying the findings in their manuscript fully available?

Reviewer #2: Yes

5. Is the manuscript presented in an intelligible fashion and written in standard English?

Reviewer #2: Yes

6. Review Comments to the Author

Reviewer #2: (No Response)

7. PLOS authors have the option to publish the peer review history of their article (what does this mean?). If published, this will include your full peer review and any attached files.

Reviewer #2: No

---

## [Editor Report · Acceptance letter]

7 Mar 2023

PONE-D-22-14999R4 

Inhibition of VEGF receptors induces pituitary apoplexy: an experimental study in mice 

Dear Dr. Hojo:

I'm pleased to inform you that your manuscript has been deemed suitable for publication in PLOS ONE. Congratulations! Your manuscript is now with our production department. 

Kind regards, 

on behalf of

Dr. Takahiro Nemoto 

Academic Editor

PLOS ONE